# *BDNF* rs962369 Is Associated with Major Depressive Disorder

**DOI:** 10.3390/biomedicines11082243

**Published:** 2023-08-10

**Authors:** Aneta Bednářová, Viera Habalová, Ivan Tkáč

**Affiliations:** 12nd Department of Psychiatry, Faculty of Medicine, Pavol Jozef Safarik University and Louis Pasteur University Hospital, 041 90 Kosice, Slovakia; 2Department of Medical Biology, Faculty of Medicine, Pavol Jozef Safarik University, 040 11 Kosice, Slovakia; viera.habalova@upjs.sk; 34th Department of Internal Medicine, Faculty of Medicine, Pavol Jozef Safarik University and Louis Pasteur University Hospital, 041 90 Kosice, Slovakia; ivan.tkac@upjs.sk

**Keywords:** schizophrenia, depression, polymorphism, *BDNF*, rs6265, rs962369

## Abstract

This study enrolled 291 patients diagnosed with depression and schizophrenia (F32, F33, and F20 according to ICD-10) and 227 ethnicity-matched control subjects. We analyzed the distribution of *BDNF* rs6265 and *BDNF* rs962369 genotypes, finding no significant associations between these and schizophrenia. We revealed a significant increase in the risk of single-episode major depression disorder (MDD) for rs962369 minor allele homozygotes (CC vs. TT+TC), an association that persisted after adjusting for age and sex (OR 3.47; 95% CI 1.36–8.85; *p* = 0.009). Furthermore, rs962369 genotype was significantly associated with an increased risk of recurrent MDD in a log-additive model (OR per C-allele 1.65; 95% CI 1.11–2.45; *p* = 0.013). A comparative analysis between MDD subtypes and between MDD subtypes and schizophrenia showed no significant differences for *BDNF* rs6265. Notably, the frequency of minor allele C of *BDNF* rs962369 varied across subgroups, with the highest frequency in patients with recurrent MDD (0.32) and the lowest in schizophrenia patients (0.20). The presence of genotypes with at least one minor allele C was significantly higher in the recurrent MDD patient group compared to the schizophrenia group. In conclusion, the *BDNF* rs962369 variant was associated with MDD but not with schizophrenia.

## 1. Introduction

Depression is a common mental disorder. Globally, it is estimated that 5% of adults suffer from the disorder [1]. Major depressive disorder (MDD) is a type of depressive disorder characterized by episodes that last at least two weeks. These episodes bring about changes in emotions, including feelings of sadness, irritation, or emptiness, and disruptions in cognition and neurovegetative functions. Such changes can impair the individual’s ability to function normally [2]. Depression rates are on the rise globally. According to the World Health Organization (WHO), the number of MDD cases grew by 18.4% from 2005 to 2015 [1], though this increase could also be attributed to population growth and/or gradually decreasing stigmatization surrounding mental illness [3]. In the first year of the COVID-19 pandemic, rates of common conditions, such as depression and anxiety, went up by more than 25% [4]. Generally, major depressive disorder (MDD) is more common in women than men. Worldwide reported female-to-male ratio can range from 1.4–2.8 to 1 [5].

Schizophrenia affects approximately 24 million people or 1 in 300 people (0.32%) worldwide. Its rate is 1 in 222 people (0.45%) among adults [6]. The female-to-male ratio of schizophrenia prevalence varies depending on the study, but it is generally thought to be around 1:1.2 to 1:1.4. This means that men are slightly more likely to develop schizophrenia than women. However, some studies did not find sex differences in the lifetime prevalence of the illness [7]. Schizophrenia is a complex genetic disorder characterized clinically by a heterogeneous syndrome comprising delusions, hallucinations, and cognitive impairment. These myriad of signs and symptoms arise from dysfunction in distributed neural circuits and complex genes-environment interactions [8,9]. In schizophrenia, primary evidence suggests symptoms of low mood, suicidal ideation, and pessimism have more specificity for depression, whereas alogia and blunted affect may have more specificity as negative symptoms. Anhedonia, anergia, and avolition may be common to both. A detailed phenomenology of schizophrenia can further distinguish depressive features from negative symptoms. However, the two domains continue to share certain phenomena, emphasizing their close relationship [10].

Schizophrenia and mood disorders overlap phenomenologically. Mood symptoms are common features of prodromal psychosis and established schizophrenia [11], whereas psychotic symptoms are often found in severe mood disorders. Such observations, as well as expanding biological evidence [12,13], call the Kraepelinian dichotomy into challenge. One area of investigation is negative symptoms. These may manifest variously as loss of motivation to act and lack of the elements that make up the normal repertoire of social and emotional response [14], but these are also commonly found in depression [15]. Both the above diagnostic entities can lead to suicidal behavior and to completed suicide.

Family and twin studies have consistently shown that both schizophrenia and depression tend to run in families, indicating a hereditary influence. These studies have estimated that the heritability of schizophrenia is around 70–80% [16,17]. Similarly, for depression, the heritability is estimated to be around 30–40% and might be higher for severe depression [18,19]. Several genes have been implicated in both schizophrenia and depression, although it is important to note that these disorders are polygenic, meaning they involve multiple genes interacting with environmental factors. Some genes associated with schizophrenia and depression include those involved in neurotransmitter systems such as dopamine (e.g., *COMT*, *DRD3*, and *DRD4*), serotonin (e.g., *SLC6A4*, *TPH1*, and *TPH2*), and glutamate (e.g., *GRIA4*)), neurodevelopmental processes (e.g., *GRIN1*), and immune system functioning (e.g., *IL6* and *IL6R*) [20,21,22,23].

The brain-derived neurotrophic factor (BDNF) protein, a member of the neuronal growth factor family, is found not only in neuronal tissues but also in nonneuronal tissues. These include endothelial cells, cardiac cells, vascular smooth muscle, leukocytes, megakaryocytes, and platelets [24]. The BDNF gene is located on chromosome 11p14.1 in humans, and it harbors a specific single nucleotide polymorphism (SNP) known as rs6265 C>T. This change results in the substitution of valine with methionine at the 66th codon (Val66Met) of the BDNF precursor, known as pro-BDNF. This substitution leads to a reduction in the secretion of the BDNF protein [25,26].

Research examining the expression of the BDNF gene and its peripheral levels in relation to major depressive disorder (MDD) has yielded inconsistent results. Some studies have found a significant association, while others have found no connection at all [27]. This inconsistency in findings is likely attributable to the genotypic and phenotypic diversity across different populations.

Several studies, along with a meta-analysis, have previously identified a higher risk for schizophrenia in individuals carrying the Met/Met variant of the non-synonymous Val66Met SNP in the BDNF gene [28]. The Val66Met (rs6265) polymorphism, a functional variation of the BDNF gene, has been extensively investigated in connection with suicidal behavior, but the results have been inconsistent. This specific SNP has been linked to an increased risk of suicidal behavior in certain ethnic groups, such as Caucasians and Asians, but this association has not been found in the overall population [29]. Additionally, another BDNF gene polymorphism, the intron variant rs962369, has been strongly associated with suicidal thoughts during antidepressant treatment [30], as well as with completed suicides [31].

Researchers are also interested in the epigenetic regulation of *BDNF*. In the *BDNF* gene, changes in DNA methylation at specific promoter regions can influence its transcriptional activity. Increased DNA methylation at the promoter regions is generally associated with decreased BDNF expression, while reduced methylation is linked to increased expression. A study by Januar et al. [32] suggested a connection between the rs6265 Val66Met polymorphism and promoter methylation, indicating that the Val66Met variant may influence the methylation status of the *BDNF* gene’s promoter region, but other studies did not confirm these findings [33,34,35].

This study aimed to evaluate the contribution of *BDNF* gene variants (rs6265, rs962369) to the susceptibility of one-episode MDD, recurrent MDD, and schizophrenia. While schizophrenia and depression are distinct conditions with different symptoms and diagnostic criteria, it was of interest to investigate whether these differences are also reflected in *BDNF* gene variant frequencies.

## 2. Materials and Methods

### 2.1. Study Sample

In total, 291 unrelated patients diagnosed with schizophrenia (F20) and major depressive disorder (MDD) (F32 and F33), as classified by the International Classification of Diseases 10th revision (ICD-10), were included in the study. The control group, comprising 227 ethnicity-matched adult volunteers, was selected to correspond with the patient group. Individuals in the control group had no known relationship to the cases nor any known psychiatric disorders or history of mental disorders.

The participants included in all groups were of Slovak origin (Caucasians). Blood samples were collected from November 2018 to January 2023 except for the period when the clinic was reprofiled for COVID-positive patients (from November 2020 to March 2021).

Participation in the testing was voluntary, and any individual could withdraw at any time during the study. The study received approval from the Ethics Committee of the Louis Pasteur University Hospital in Kosice, and all subjects provided their written informed consent. The research was conducted in accordance with the principles of the Declaration of Helsinki.

### 2.2. Genotyping

DNA from peripheral blood was extracted using the QIAamp DNA Blood Mini QIAcube Kit according to the manufacturer’s instructions on the QIAcube—robotic workstation for automated purification of DNA, RNA, or proteins (QIAGEN, Hilden, Germany). Genotyping of the *BDNF* rs6265 and *BDNF* rs962369 was performed using asymmetric real-time polymerase chain reaction (primers ratio 1:10). The high-resolution melting analysis was conducted in the presence of an unlabeled probe using the Eco™ Real-Time PCR System (Illumina, Inc., San Diego, CA, USA). The reaction mixture was composed of 1× MeltDoctor™ HRM Master Mix (Applied Biosystems by Thermo Fisher Scientific, Vilnus, Lithuania), the appropriate oligonucleotides, and 20 ng of template DNA, all in a final volume of 15 μL. Genotypes were identified using the Eco™ Software 4.1, and the oligonucleotides were specifically designed in our laboratory (as shown in Table 1).

### 2.3. Statistical Analysis

The Hardy-Weinberg equilibrium (HWE) was assessed for the tested groups by comparing the observed numbers of each genotype with those expected under the HWE for the estimated allele frequency. A comparison of frequencies was made using χ^2^ -test. Multivariable logistic regression was used to study the association between diseases and genotypes and for adjustment for covariates or stratification. Results are expressed as odds ratios (OR) and 95% confidence intervals (CI). Online available software SNPStats (accessed on 20 February 2018) was used for statistical analysis [36]. The association between gene variations and phenotypes was analyzed using several genetic models, including codominant, dominant, overdominant, recessive, and log-additive. The analysis of this association was conducted through binomial logistic regression. Because two SNPs were evaluated in the present study, Bonferroni’s correction was applied, and *p*-value of <0.025 was considered statistically significant.

## 3. Results

### 3.1. Characteristics of Study Groups

Two hundred and ninety-one patients diagnosed with depression and schizophrenia (F32, F33, and F20 according to ICD-10) and 227 control subjects were included in the present study. Sex and age characteristics according to the diagnoses can be found in Table 2. The basic information on the two SNPs examined in this study is summarized in Table 3. The *BDNF* rs6265 and *BDNF* rs962369 genotype distributions were in the Hardy–Weinberg equilibrium in controls (χ^2^-test; *p* = 0.42 resp. *p* = 0.19).

The sex ratios of patients with major depressive disorders and schizophrenia in our study are consistent with global epidemiology. The female-to-male ratio for major depressive disorders (F32 and F33) was 2.3:1, while the male-to-female ratio for schizophrenia was 1.7:1.

### 3.2. Associations between BDNF Variants and Different Types of Mental Disorders

The distribution of *BDNF* rs6265 genotypes (Figure 1) and alleles was similar between controls and patients with recurrent MDD (F33) or schizophrenia (F20) (Table 4, Figure 1). The crude analysis showed a higher frequency of the minor homozygote genotype TT (MetMet) in patients with one-episode MDD (F32) than in controls under the recessive genetic model. However, this difference did not remain statistically significant after adjusting for sex and age or after stratifying by sex.

The analysis of the second *BDNF* genetic variant rs962369 (Table 4, Figure 2) did not confirm its association with schizophrenia in any of the five genetic models nor after sex stratification. We have revealed a significantly increased risk of one-episode MDD for the minor allele homozygotes (CC vs. TT+TC). This association persisted after adjustment for age and sex. Further, we analyzed the association between selected SNP and recurrent MDD risk. Genotypes were significantly associated with increased recurrence of MDD risk in the log-additive model.

### 3.3. Comparison of Different Types of Mental Disorders in Relation to BDNF Variants

To examine whether the analyzed mental disorders differ in their genetic overlap, we performed a comparative analysis between MDD subtypes, and we also compared *BDNF* variants between MDD subtypes and schizophrenia. When comparing all patient groups to each other for *BDNF* rs6265, no statistically significant differences in genotype or allele frequency were observed.

The frequency of minor allele C varied across subgroups of patients for *BDNF* rs962369. The highest frequency of the allele C was observed in patients with recurrent MDD (0.32) and the lowest in patients with schizophrenia (0.20). The presence of genotypes with at least one minor allele C (genotypes TC+CC) was significantly higher in the group of patients with recurrent MDD in comparison with a group of patients with schizophrenia (55.7% vs. 36.4%; *p* = 0.009). No statistically significant difference was observed in genotype distribution or allele frequency between patients with one-episode MDD and patients with schizophrenia.

## 4. Discussion

The most important finding of the present study was that *BDNF* intron variant, rs962369 was associated with a significant 3.5-times increase in the risk of single-episode MDD for subjects who were homozygous for the minor allele C. In addition, the variant rs962369 demonstrated an effect on the development of recurrent MDD, most significantly in the log-additive model (1.7-times per C-allele).

The *BDNF* rs962369 variant showed differences across subgroups of patients. The presence of at least one minor allele (TC+CC genotypes) was significantly higher in the group of patients with recurrent MDD compared to the group of patients with one-episode MDD and the group of patients with schizophrenia, respectively. We did not come across any comparative analyses in the literature that specifically investigated the genetic *BDNF* variants in relation to selected mental disorders in our study, namely MDD or schizophrenia. Genetic variant rs962369 is less widely referenced or studied and is mainly associated with attempted suicide [31] or suicidal ideation [30].

Suicidal ideation and completed suicide can occur in both one-episode and recurrent depression, but the risk is generally considered higher in individuals with recurrent depression due to the chronic and repetitive nature of their condition. Our recently published study [26] demonstrated a significant association between rs962369 and completed suicide. Based on our findings, we hypothesize that rs962369 is one of the risk factors that contribute to the development of recurrent MDD and may ultimately lead to completed suicide. We observed no statistically significant differences in *BDNF* rs6265 (Val66Met) genotype or allele frequency between all patient groups and controls after adjusting for sex and age. Likewise, no association between depression status and the Val66Met polymorphism was confirmed, neither in a retrospective study on 7389 British study subjects participating in the European Prospective Investigation [37] nor in a meta-analysis involving 14 studies with a total sample size of 2812 cases with DSM-III or -IV defined MDD and 10,843 non-depressed controls [38].

Similarly, the association was not observed in the onset of depression in European (white) premenopausal women [39] or in the Russian study of newly admitted depressed patients. However, the already-mentioned study from Russia showed an association between the BDNF gene variant rs6265 and the severity of depression. In contrast, an increased risk of developing MDD for homozygote genotype TT (Met/Met) and the possible role of *BDNF* in the pathogenesis of major depression was observed in studies from Asia [40,41].

The results from studies investigating the *BDNF* rs6265 gene variant and its relationship to the onset of schizophrenia were inconclusive. In a small Greek study [42] and several meta-analyses conducted in 2006, 2007 [43], and 2016 [28], it was observed that homozygous carriers of the Met allele were overrepresented in the group of patients with schizophrenia compared to the control group. However, other studies, such as those conducted on the Chinese Han population [44] or individuals of Slavic descent [45,46], did not find any association between the Val66Met polymorphism and schizophrenia. Our data also support the latter findings. Furthermore, a recently published meta-analysis that included 11,480 patients with schizophrenia and 13,490 healthy controls confirmed the lack of association of Val66Met polymorphism with schizophrenia [47].

MDD is a multifactorial disorder, and genetics is just one of the factors that have an additive effect leading to the disorder. It is important to note that while *BDNF* has shown associations with various psychiatric disorders, it is not a definitive diagnostic marker on its own. Psychiatric diagnoses rely on a comprehensive assessment that includes clinical symptoms, medical history, and other relevant factors. *BDNF* can serve as a potential biological marker that, when considered alongside other clinical information, may contribute to the differential diagnosis process or identification of patients requiring increased attention. However, its use in clinical practice is still an area of ongoing research, and further studies are needed to establish its diagnostic utility.

The main limitation of the present study is a relatively small group of included participants. In genetic association studies with smaller number of included subjects, it is harder to spot small genetic effects, leading to possibly missed findings. Different groups of people can have different genetic variations, thus the link between certain genes and diseases might differ among ethnicities. Therefore, if a study looks at diverse ethnic groups, the findings might change. Conditions like MDD and schizophrenia are complicated and can have various root causes and ways they show up in people. Some specific versions of these disorders, or particular symptoms, might have a stronger genetic link. The different results we see now could be due to factors such as gender, environment, and the presence of multiple genes or how they interact, which we still need to explore. In addition, the mental disorder may also manifest itself in patients from the control group in the future.

## Figures and Tables

**Figure 1 biomedicines-11-02243-f001:**
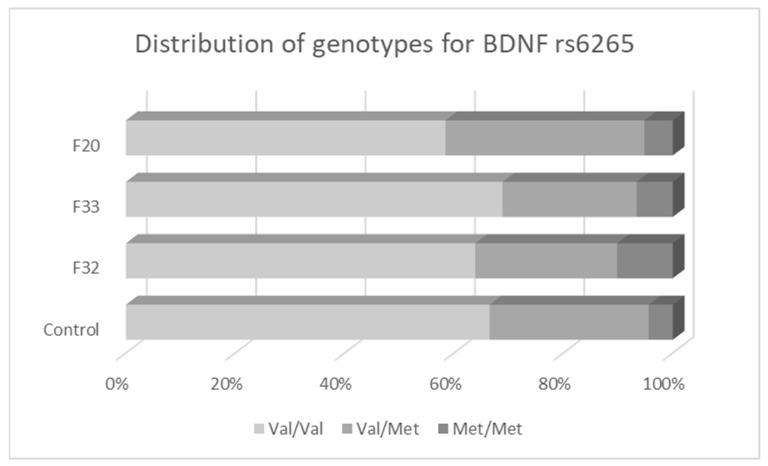
Distribution of genotypes for *BDNF* variant rs6265.

**Figure 2 biomedicines-11-02243-f002:**
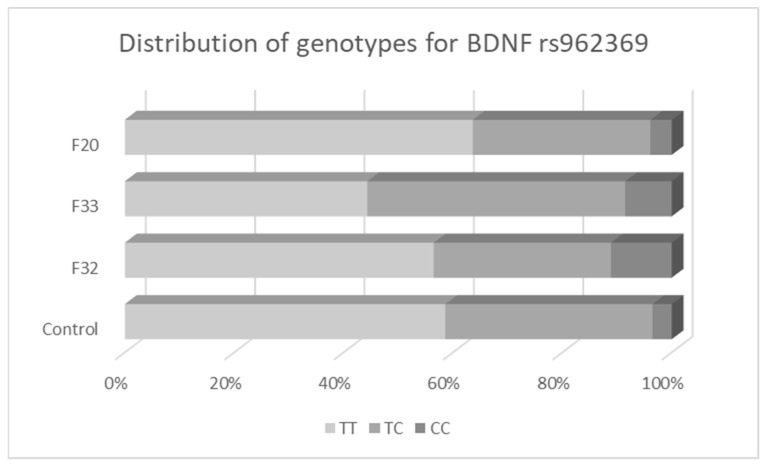
Distribution of genotypes for *BDNF* variant rs962369.

**Table 1 biomedicines-11-02243-t001:** Sequences of oligonucleotides.

Gene	Oligonucleotides	Sequences
***BDNF* rs6265**	forward-limit	5′-GCCGAACTTTCTGGTCCTCATCC-3′
	reverse-excess	5′-AAGGCAGGTTCAAGAGGCTTG-3′
	probe	5′-GCTCTTCTATCACGTGTTCGAAAGTGTC-Phos
***BDNF* rs962369**	forward-limit	5′-GACATTTTTATGAGAAGGGTTTACATAAG-3′
	reverse-excess	5′-AAAGAATTGCTCACTGTAATGAC-3′
	probe	5′-TGCCAAGAGAGTTGAGTCCATGG-Phos

**Table 2 biomedicines-11-02243-t002:** Age and gender characteristics of the studied groups.

Variables		MDD, One Episode F32	MDD, Recurrent F33	Schizophrenia F20	Controls
Total		108	106	77	227
Sex	Females	69 (64%)	80 (75%)	28 (36%)	115 (51%)
	Males	39 (36%)	26 (25%)	49 (63%)	112 (49%)
Age	Mean ± SD	48.55 ± 15.72	57.04 ± 10.48	44.71 ± 13.39	54.06 ± 22.55

**Table 3 biomedicines-11-02243-t003:** Genotype and alleles distribution of *BDNF* rs6265 (Val66Met) and rs962369 variants in groups of patients with different types of psychiatric disorders.

		*BDNF* rs6265					*BDNF* rs962369				
		Genotypes			Alleles Frequency		Genotypes			Alleles Frequency	
		CC (%)	CT (%)	TT (%)	C (Val)	T (Met)	TT (%)	TC (%)	CC (%)	T	C
**Controls**											
Total	227	151 (66.5)	66 (29.1)	10 (4.4)	0.81	0.19	133 (58.6)	86 (37.9)	8 (3.5)	0.78	0.22
Females	115	72 (62.6)	36 (31.3)	7 (6.1)	0.78	0.22	64 (55.6)	46 (40.0)	5 (4.3)	0.76	0.24
Males	112	79 (70.5)	30 (26.8)	3 (2.7)	0.84	0.16	69 (61.6)	40 (35.7)	3 (2.7)	0.79	0.21
**F32**											
Total	108	69 (63.9)	28 (25.9)	11 (10.2)	0.77	0.23	61 (56.5)	35 (32.4)	12 (11.1)	0.73	0.27
Females	69	45 (65.2)	17 (24.6)	7 (10.2)	0.78	0.22	37 (53.6)	26 (37.7)	6 (8.7)	0.72	0.28
Males	39	24 (61.5)	11 (28.2)	4 (10.3)	0.76	0.24	24 (61.5)	9 (23.1)	6 (15.4)	0.73	0.27
**F33**											
Total	106	73 (68.9)	26 (24.5)	7 (6.6)	0.81	0.19	47 (44.3)	50 (47.2)	9 (8.5)	0.68	0.32
Females	80	55 (68.8)	20 (25.0)	5 (6.3)	0.81	0.19	38 (47.5)	34 (42.5)	8 (10.0)	0.69	0.31
Males	26	18 (69.2)	6 (23.1)	2 (7.7)	0.81	0.19	9 (34.6)	16 (61.5)	1 (3.8)	0.65	0.35
**F20**											
Total	77	45 (58.4)	28 (36.4)	4 (5.2)	0.77	0.23	49 (63.6)	25 (32.5)	3 (3.9)	0.80	0.20
Females	28	18 (64.3)	8 (28.6)	2 (7.1)	0.79	0.21	20 (71.4)	6 (21.4)	2 (7.1)	0.82	0.18
Males	49	27 (55.1)	20 (40.8)	2 (4.1)	0.76	0.24	29 (59.2)	19 (38.8)	1 (2.0)	0.79	0.21

**Table 4 biomedicines-11-02243-t004:** Association analysis of *BDNF* rs6265 and rs962369 variants with susceptibility to different types of psychiatric disorders.

	Disorder	Genetic Model	Crude Analysis		Sex and Age-Adjusted Analysis	
***BDNF* rs6265**			**OR (95% C.I.)**	***p* (Wald Test)**	**OR (95% C.I.)**	***p* (Wald Test)**
	**F32**	recessive (TT vs. CC+CT)	2.46 (1.01–5.99)	0.049	2.37 (0.96–5.89)	0.063
	**F33**	all tested models	non-significant	>0.05	non-significant	>0.05
	**F20**	all tested models	non-significant	>0.05	non-significant	>0.05
***BDNF* rs962369**						
	**F32**	recessive (CC vs. TT+TC)	3.42 (1.36–8.64)	0.009	3.47 (1.36–8.85)	0.009
	**F33**	log-additive (per C allele)	1.71 (1.16–2.51)	0.006	1.65 (1.11–2.45)	0.013
	**F20**	all tested models	non-significant	>0.05	non-significant	>0.05

## Data Availability

The data that support the findings of this study are available upon request.

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
