# Peer review of "BDNF rs962369 Is Associated with Major Depressive Disorder"

_biomedicines, 2023, doi:10.3390/biomedicines11082243_

Round 1
Reviewer 1 Report
Here are a few suggestions:
1. Please include this reference on page 2, lines 65-67 "Some genes associated with schizophrenia and depression..... immune system functioning", and include some examples.
2. Include this reference in the manuscript (DOI: 10.1038/tp.2015.114.). Please do mention that rs962369 is present in the BDNF promoter I and is a methylation site and is associated with later-life depression. rs 6265 is associated with the incidence of suicidal activity and found in BDNF promoter IV.
3. You may also include this reference in the manuscript (
DOI: 10.1111/acer.13414).
Author Response
-

Reviewer 2 Report
In their short communication, Bednářová, Habalová and Tkáč studied an association of two BDNF polymorphisms with schizophrenia and MDD, two mental disorders that share come common signs. They failed to reveal a significant link to the pathologies for BDNF rs6265, while established a significant association between the other studied SNP, BDNF rs962369, and MDD, but not schizophrenia. The study is a modest, but valuable contribution to the field and would help in practical issues of a known Kraepelinian dichotomy. In my opinion, the manuscript represents interesting and novels facts with a good citing potential. Thus, I recommend to accept the manuscript with the following minor suggestions (with reference to line numbers in the PDF file):
1. Line 53: ‘Kraepelinian dichotomy’ is a better and more widely used term than ‘Kraepelin's dichotomy.’
2. Line 214: ‘…14 studies with a total sample size [28].’ – the phrase looks as trunkated. I can guess that it should have state the size number, for example, ‘with a total sample size of 5’000 subjects’, but the size number has been mistakingly missed.
3. The authors reported a significant increase in the risk of single-episode MDD for rs962369 minor allele homozygotes (CC vs. TT+TC), whereas the same SNP was significantly associated with an increased risk of recurrent MDD in a log-additive model, i.e. TT vs. CT vs. CC. Thus, CC was the most different genotype for one-episode MDD and TT for recurrent. Both findings are not contradictory, but why don’t the authors consolidate the two samples into a common F32+F33 combined sample with an addition of the relevant lines/columns to Tables 2, 3 and 4? The sample would become larger and the result become more powerful statistically.
4. In Table 2, male/female ratio for schizophrenia cases was 63% men to 36% women, while for one-episode MDD it was opposite, 36% to 64%, and for recurrent MDD, even 25% men to 75% women. Is it normal for there disorders? Please, indicate in the text, if the raios are consistent with the global epidemiology, and if not, what can be the cause of the bias.
5. In lines 154-155 and some other places, age and sex adjustment and sex stratification is mentioned. Please, specify in paragraph 2.3 ‘Statistical analysis’ the method(s) used for the adjustment and stratification. Also, the name and version of statistical software should be indicated.
6. In ‘Introduction’, the authors declared that ‘In the first year of the COVID-19 pandemic, rates of common conditions such as depression and anxiety went up by more than 25 % [4].’ (lines 36-38). Nothing is said in the text for covid effect on the schizophrenia rates, though I could guess they also increased. Notably, line 103 reports that ‘samples were collected from November 2018 to January 2023.’ It means that the sampling occurred during the pre-covid, covid, and post-covid periods. I am sure that it must be addressed in the revised manuscript, if there was any difference or irregularity in sampling during the covid and non-covid times. For instance, a reader can suspect that the F20 samples were collected before the covid, while MDD samples were pick up during/after covid, and that’s why the study revealed no link to schizophrenia, but a significant link to F32&F33, etc.
Thus, I recommend to accept the manuscript with the above-mentioned minor revisions.
Author Response
In their short communication, Bednářová, Habalová and Tkáč studied an association of two BDNF polymorphisms with schizophrenia and MDD, two mental disorders that share come common signs. They failed to reveal a significant link to the pathologies for BDNF rs6265, while established a significant association between the other studied SNP, BDNF rs962369, and MDD, but not schizophrenia. The study is a modest, but valuable contribution to the field and would help in practical issues of a known Kraepelinian dichotomy. In my opinion, the manuscript represents interesting and novels facts with a good citing potential. Thus, I recommend to accept the manuscript with the following minor suggestions (with reference to line numbers in the PDF file):
Comment 1. Line 53: ‘Kraepelinian dichotomy’ is a better and more widely used term than ‘Kraepelin's dichotomy.’
Response 1. We changed the text according to the reviewer’s recommendation.
Comment 2. Line 214: ‘…14 studies with a total sample size [28].’ – the phrase looks as trunkated. I can guess that it should have state the size number, for example, ‘with a total sample size of 5’000 subjects’, but the size number has been mistakingly missed.
Response 2. Thank you for pointing out this mistake. We inserted the end of the sentence: “2812 cases with DSM-III or -IV defined MDD and 10 843 nondepressed controls.“
Comment 3. The authors reported a significant increase in the risk of single-episode MDD for rs962369 minor allele homozygotes (CC vs. TT+TC), whereas the same SNP was significantly associated with an increased risk of recurrent MDD in a log-additive model, i.e. TT vs. CT vs. CC. Thus, CC was the most different genotype for one-episode MDD and TT for recurrent. Both findings are not contradictory, but why don’t the authors consolidate the two samples into a common F32+F33 combined sample with an addition of the relevant lines/columns to Tables 2, 3 and 4? The sample would become larger and the result become more powerful statistically.
Response 3. Log-additive model for rs962369 reflects increased risk per one C allele. We inserted this information into Table 4. Thus, in both models (recessive and log-additive), the C allele reflected an increased risk of developing depression and despite a lower number of subjects in each group, the results were statistically significant. We decided to analyze patients with F32 and F33 diagnoses separately because these diagnoses are more specific than the general diagnosis of depression. This allowed us to create albeit smaller but more homogeneous groups of patients, which made it easier to identify the effects of the BDNF variants. Our decision was inspired by an article: Flint J. The genetic basis of major depressive disorder. Mol Psychiatry. 2023 Jan 26. doi: 10.1038/s41380-023-01957-9. Epub ahead of print. PMID: 36702864.
Comment 4. In Table 2, male/female ratio for schizophrenia cases was 63% men to 36% women, while for one-episode MDD it was opposite, 36% to 64%, and for recurrent MDD, even 25% men to 75% women. Is it normal for there disorders? Please, indicate in the text, if the raios are consistent with the global epidemiology, and if not, what can be the cause of the bias.
Response 4. Due to the small number of individuals in our study, we did not focus on the relative representation of men and women within the diagnostic groups. Generally, major depressive disorder (MDD) is more common in women than men. From the literature as well as from clinical practice we know that some factors may contribute to the higher prevalence of MDD in women, including biological or social factors. Life events, such as pregnancy, fertility and menopause, are unique to women and can affect the onset and course of mental disorders. The female sex and reproductive transitions impact various aspects of brain biology and pathobiology. These biological differences can impact differential gene expression or organ development (Cyranowski et. al 2000; Li et al, 2022). Social factors like experiencing stressful life events, e.g. sexual assault and/or intimate partner violence can also increase the risk of MDD. Women are more likely to report feeling isolated and unsupported (Gutiérrez-Rojas et al., 2020).
Li X, Zhou W, Yi Z. A glimpse of gender differences in schizophrenia. Gen Psychiatr. 2022 Aug 30;35(4):e100823. doi: 10.1136/gpsych-2022-100823. PMID: 36118418; PMCID: PMC9438004
Cyranowski JM, Frank E, Young E, Shear MK. Adolescent Onset of the Gender Difference in Lifetime Rates of Major Depression: A Theoretical Model. Arch Gen Psychiatry. 2000;57(1):21–27. doi:10.1001/archpsyc.57.1.21).
Gutiérrez-Rojas L, Porras-Segovia A, Dunne H, Andrade-González N, Cervilla JA. Prevalence and correlates of major depressive disorder: a systematic review. Braz J Psychiatry. 2020 Nov-Dec;42(6):657-672. doi: 10.1590/1516-4446-2020-0650. PMID: 32756809; PMCID: PMC7678895.
We have added to the Introduction: Generally, major depressive disorder (MDD) is more common in women than men. Worldwide reported female to male ratio can range from 1.4 - 2.8 to 1 (Gutiérrez-Rojas et al., 2020).
Several psychosocial factors can contribute to the development of schizophrenia, including loneliness, adverse life events and lack of family/social support. Men are more likely to be diagnosed with schizophrenia later in life, which may lead to a longer duration of untreated illness.
We have added to the Introduction: The female to male ratio of schizophrenia prevalence varies depending on the study, but it is generally thought to be around 1:1.2 to 1:1.4. This means that men are more likely to develop schizophrenia than women. However, some studies did not find sex differences in the lifetime prevalence of the illness (Aleman et al., 2003).
Aleman A, Kahn RS, Selten J-P. Sex differences in the risk of schizophrenia: evidence from meta-analysis. Arch Gen Psychiatry 2003;60:565–71. 10.1001/archpsyc.60.6.565
and we also added to the Results: The sex ratios of patients with major depressive disorders and schizophrenia in our study are consistent with global epidemiology. The female to male ratio for major depressive disorders (F32 and F33) was 2.3:1, while the male to female ratio for schizophrenia was 1.7:1.
Comment 5. In lines 154-155 and some other places, age and sex adjustment and sex stratification is mentioned. Please, specify in paragraph 2.3 ‘Statistical analysis’ the method(s) used for the adjustment and stratification. Also, the name and version of statistical software should be indicated.
Response 5. “Multivariable logistic regression was used to study the association between diseases and genotypes and for adjustment for covariates or stratification. Results are expressed as odds ratios (OR) and 95% confidence intervals (CI). Online software SNP-stats was used for statistical analysis.” This statement is now included in paragraph 2.3.
Comment 6. In ‘Introduction’, the authors declared that ‘In the first year of the COVID-19 pandemic, rates of common conditions such as depression and anxiety went up by more than 25 % [4].’ (lines 36-38). Nothing is said in the text for covid effect on the schizophrenia rates, though I could guess they also increased. Notably, line 103 reports that ‘samples were collected from November 2018 to January 2023 .’ It means that the sampling occurred during the pre-covid, covid, and post-covid periods. I am sure that it must be addressed in the revised manuscript, if there was any difference or irregularity in sampling during the covid and non-covid times. For instance, a reader can suspect that the F20 samples were collected before the covid, while MDD samples were pick up during/after covid, and that’s why the study revealed no link to schizophrenia, but a significant link to F32&F33, etc.
Response 6: All samples were taken in the pre- and post-COVID periods. During the COVID period, our psychiatric clinic was reprofiled and the collection of biological material for scientific purposes was suspended by the hospital management, so patient recruitment was not ongoing.
We corrected the text in paragraph 2.1 Study sample: ‘samples were collected from November 2018 to January 2023 except for the period when the clinic was reprofiled for COVID-positive patients (from November 2020 to March 2021).
We are grateful for the excellent comments that contributed to the improvement of the manuscript.